# Second-Generation Antipsychotics’ Effectiveness and Tolerability: A Review of Real-World Studies in Patients with Schizophrenia and Related Disorders

**DOI:** 10.3390/jcm11154530

**Published:** 2022-08-03

**Authors:** Michele Fabrazzo, Salvatore Cipolla, Alessio Camerlengo, Francesco Perris, Francesco Catapano

**Affiliations:** Department of Psychiatry, University of Campania “Luigi Vanvitelli”, Largo Madonna Delle Grazie 1, 80138 Naples, Italy; salvatore2211@gmail.com (S.C.); alessiocamerlengo90@gmail.com (A.C.); francesco.perris@unicampania.it (F.P.); francesco.catapano@unicampania.it (F.C.)

**Keywords:** schizophrenia, negative symptoms, real-world studies, real-world effectiveness, tolerability, treatment adherence, second-generation antipsychotics, long-acting injectable antipsychotics

## Abstract

Despite methodological limitations, real-world studies might support clinicians by broadening the knowledge of antipsychotics’ (APs) effectiveness and tolerability in different clinical scenarios and complement clinical trials. We conducted an extensive literature search in the PubMed database to evaluate the effectiveness and tolerability profiles of second-generation antipsychotics (SGAs) from real-world studies to aid clinicians and researchers in selecting the proper treatment for patients with schizophrenia and related disorders. The present review evidenced that SGAs demonstrated superior effectiveness over first-generation antipsychotics (FGAs) in relapse-free survival and psychiatric hospitalization rate and for treating negative symptoms. Persistence and adherence to therapy were higher in SGAs than FGAs. Most studies concluded that switching to long-acting injectables (LAIs) was significantly associated with a lower treatment failure rate than monotherapy with oral SGAs. Considerable improvements in general functionality, subjective well-being, and total score on global satisfaction tests, besides improved personal and social performance, were reported in some studies on patients treated with LAI SGAs. Clozapine was also associated with the lowest rates of treatment failure and greater effectiveness over the other SGAs, although with more severe side effects. Effectiveness on primary negative symptoms and cognitive deficits was rarely measured in these studies. Based on the data analyzed in the present review, new treatments are needed with better tolerability and improved effectiveness for negative, affective, and cognitive symptoms.

## 1. Introduction

Schizophrenia is among the most disabling mental health conditions [1] and affects approximately 24 million people worldwide [2,3]. In addition, subjects affected by schizophrenia and related disorders have a 10–25-year reduction in life expectancy than the general population due to the increased rates of comorbid physical illnesses, smoking, and substance abuse, rates of suicide as common causes of death, and reduced health-seeking behavior [4,5,6].

Patients with schizophrenia and related disorders may experience positive, negative, affective, and cognitive symptoms influencing many aspects of their daily functioning [7,8,9,10,11,12,13].

The psychopharmacological treatment of schizophrenia and other psychotic disorders relies mainly on antipsychotics (APs), which are traditionally divided into two classes: first-generation antipsychotics (FGAs) and second-generation antipsychotics (SGAs) [14,15,16]. Both classes of drugs are effective in relieving the positive symptoms of schizophrenia. Instead, evidence of the efficacy on negative, affective, and cognitive symptoms is inconclusive, and these dimensions remain the unmet needs of schizophrenia treatment [17,18,19].

APs may also induce different side-effect profiles [20], occasionally perceived by patients as distressing and disabling [21]. In general, side effects include extrapyramidal side effects (EPS), increased prolactin plasma levels, metabolic complications such as weight gain, metabolic syndrome, hyperlipidemia, and type 2 diabetes, which may reduce life expectancy [22,23,24]. Specifically, FGAs might induce hyperprolactinemia and frequent adverse motor effects, such as EPS, as well as increasing disability and stigma related to the disease [24].

SGAs are associated, although not consistently [25,26,27], with a reduced incidence of EPS, compared to FGAs, with a few distinctions between both medications [28,29]. However, the difference between the two classes of APs is clinically relevant, as EPSs are associated with reduced treatment adherence, depression, suicide, secondary negative symptoms, worse cognitive performance, deficits in motor skills and verbal learning, attention, and working memory [30,31,32,33]. Furthermore, EPSs often require additional treatment with anticholinergic drugs, burdening patients with adverse effects such as memory impairment, delirium, and autonomic nervous system dysfunctions.

APs may prove to be ineffective for many patients [34]. In addition, a few of them experience at least one relapse over the five years after the beginning of therapy [35]. Between a quarter and a third of affected patients manifest treatment resistance, and only 17.5% might respond to clozapine [34,36]. Therefore, a key component of the long-term management of schizophrenia and related disorders is to select an appropriate antipsychotic treatment for the needs of each individual [37,38]. The efficacy and tolerability of antipsychotic treatment might profoundly affect adherence to therapy and clinical response, with the risk of relapses [39,40].

Adverse effects are also a frequent cause for discontinued treatment, besides lack of insight, disease severity, and treatment characteristics. In addition, adverse effects may impact environmental factors such as patient’s erroneous belief in the effectiveness of medication, and substance abuse [39]. For this reason, there is a need for new treatments with improved tolerability and efficacy for negative, affective, and cognitive symptoms.

In the last 15 years, some studies have investigated the effectiveness of SGAs compared to FGAs for schizophrenia and related disorders, leading to reconceiving trials’ design using APs, as in the US Clinical Antipsychotic Trials of Intervention Effectiveness (CATIE) [26] and the UK Cost Utility of the Latest Antipsychotic Drugs in Schizophrenia Study (CUtLASS) [17]. The two trials measured short- and mid-term outcomes, not always considering the real-world clinical practice and outcome measures besides positive symptoms (e.g., exclusion of comorbidity with substance abuse, predominance of chronic patients, and lack of quality of life/well-being measures) [40]. Furthermore, the European First Episode Schizophrenia Trial (EUFEST) compared the effectiveness of some SGAs with that of a low dose of haloperidol in first-episode schizophrenia at 1-year follow-up. SGAs were associated with a higher retention rate than haloperidol (primary outcome). However, the psychopathological scores’ mean reduction did not vary [41]. A secondary analysis showed that most SGAs had higher response and remission rates than haloperidol [42]. All treatment groups were associated with worsened hypertriglyceridemia or hyperglycemia. Only ziprasidone was less associated with weight gain [43]. These results disagreed with those reported in a chart review demonstrating that SGAs in first-episode patients had a three times higher incidence of metabolic syndrome with respect to FGAs [44]. However, the study had a longer follow-up period (3 years) than the EUFEST trial. Overall, the available evidence does not coherently indicate superior effectiveness and tolerability for SGAs.

One of the most considerable challenges in treating patients with schizophrenia and related disorders is the life-long functional disability associated with negative symptoms, cognitive impairment, and increased treatment resistance after each acute episode. Consequently, the primary goal of antipsychotic treatment should be not only to achieve a partial (or optimal) remission of symptoms in the acute phase but also to improve long-term outcomes and reduce the risk of secondary negative symptoms and worsening of cognitive impairment [45,46].

Harmonizing the results of randomized clinical trials (RCTs) with those of observational studies remains a challenge for clinical medicine. Although RCTs are considered the “gold standard” for evaluating the efficacy and safety of an intervention, observational studies conducted in a real-world scenario help provide evidence of the intervention in clinical practice effectiveness. Ref. [47], indeed, reported that “real-world effectiveness” is one of the last five years’ significant research trends [47].

For a clinician, assessing both efficacy and effectiveness remains a crucial factor. Indeed, observational studies are beneficial in clinical situations rarely tested in RCTs and provide reliable real-world evidence. Specifically, RCTs evaluate interventions under ideal conditions in highly selected populations, whereas observational studies examine effects in naturalistic settings. Furthermore, RCTs results might not apply to the entire population of patients due to complex clinical presentations and poor responses to standard treatments in “real-world” settings.

On the other hand, dissimilar findings may arise due to such issues as selection bias, confounding, statistical power, and differential adherence and follow-up. Furthermore, real-world studies encompass a wide range of research methods and data sources and can be broadly categorized as non-interventional studies, patient registries, claims database studies, patient surveys, and electronic health record studies. Real-world studies can also be categorized into prospective studies, which generally require primary data collection, and retrospective studies, which use secondary data gathered over a long period (i.e., data initially collected for other purposes). Nevertheless, a recent Cochrane review showed little evidence that the results of observational studies and RCTs are systematically discordant [48]. Thus, studies on clinical effectiveness and naturalistic outcomes cannot replace RCTs, which remain complementary and fundamental to gathering helpful information.

This review aims to provide an update of the primary therapeutic and side-effect profiles of SGAs, focusing on real-world studies to enable clinicians and researchers to select the most appropriate treatment for adult patients ≥ 18 years diagnosed with schizophrenia or related disorders.

## 2. Methods

We conducted an extensive literature search in the PubMed database from inception until May 2022, with English as a language filter. This review was conducted according to the Preferred Reporting Items for Systematic Review and Meta-Analysis (PRISMA) statement, as applicable [49]. The search was conducted with the following terms (MeSH headings): ((“Adult”[Mesh]) AND (“Humans”[Mesh]) AND (“Real-World”) AND ((“Schizophrenia”[Mesh]) OR (“Schizophrenia Spectrum and Other Psychotic Disorders”[Mesh])) AND ((“Antipsychotic Agents/adverse effects”[Mesh]) OR (“Antipsychotic Agents/therapeutic use”[Mesh])) NOT (“Electroconvulsive Therapy”[Mesh]) NOT (“Transcranial Magnetic Stimulation”[Mesh])). In addition, we hand-searched the reference lists of included articles of any study on our topic of interest.

We focused on real-world studies, including prevalently longitudinal comparative studies (i.e., cohort or case–control studies). We identified schizophrenia and/or schizophrenia spectrum and other psychotic disorders as the mental disorders of interest for the scoping review, including only studies on psychopathological symptoms assessment through standardized rating scales. Furthermore, we included studies on patients treated with SGAs, or co-treated with FGAs and SGAs in the oral or long-acting injectable (LAI) formulations. Specifically, we selected studies containing data on individual drugs or grouped SGAs that reported the effectiveness and tolerability outcomes for adult participants ≥18 years. Moreover, we included studies evaluating both effectiveness and/or tolerability in patients switching from oral SGAs or FGAs to LAI SGAs.

The primary outcomes of interest were the effectiveness of oral and/or LAI formulations of SGAs on positive, negative, affective, and cognitive symptoms and their tolerability profile. In particular, we considered of interest studies reporting one or more of the following elements: (1) ≤20% reduction on the psychopathology assessment scale (i.e., BPRS); (2) improvements in quality of life rated by specific scales (i.e., Subjective Well-Being under Neuroleptics Scale [SWN-S] and the Treatment Satisfaction Questionnaire for Medication [TSQM]); (3) magnitude of treatment effects on severity measures (i.e., the Positive and Negative Syndrome Scale [PANSS], the Brief Psychiatric Rating Scale [BPRS], the Clinical Global Impression—Severity scale [CGI-S], and Quality of Life [QoL] scores); (4) improvement in negative symptoms; (5) effects on cognitive performance (evaluated by standard neuropsychological instruments); (6) improvement in global and social functioning, self-care, and disturbing/aggressive behavior (i.e., evaluated by the Global Assessment of Functioning [GAF] or the Personal and Social Performance [PSP] scale scores or defined as an increase in at least one activity in which the patient participated, compared to the baseline activity); (7) assessment of rate and time to treatment discontinuation, defined as stopping the AP medication started in baseline conditions and/or adding a new AP; (8) persistence/compliance/adherence on medications (measured as pill counts, pharmacy records, and proportion of adherent/non-adherent patients); (9) occurrence of any mental health events (suicide, hospitalization, or emergency department visits); (10) risk of rehospitalization and treatment failure (suicide attempt, discontinuation or switch to other medications, or death).

In addition, we considered of interest studies reporting any new onset or worsening side effects, i.e., EPS, hyperprolactinemia, diabetes, ketoacidosis, hyperglycemic state, weight gain/overweight/obesity, hyperlipidemia or hypercholesterolemia, hypertriglyceridemia, hypertension, and metabolic syndrome. We considered suitable and recorded any definition of these clinical entities, including diagnoses based on any coding system (e.g., ICD-10) and exposure to specific treatments (e.g., antihypertensives).

We excluded studies on pregnant women and considered only studies containing results on at least one outcome of interest (effectiveness or tolerability, or both).

S.C. and A.C. extracted the relevant data, and synthesized them in a tabular format; F.M., F.P. and F.C. triple-checked the extracted data for accuracy; M.F., S.C. and A.C extracted the data on study characteristics (type of study, number of participants/sample size, and psychopathological diagnostic tools), outcome measures (proportion of patients with schizophrenia and related disorders, psychopathological assessment tools used to evaluate the severity of disease), and therapeutic intervention types (oral vs. LAI SGAs).

Two authors of the present review (S.C. and A.C.) independently assessed the quality and risk of bias in the non-randomized studies of interventions (NRSIs) included in the present review through the ROBINS-I tool (Risk of Bias in Non-randomized Studies of Interventions). Such a tool [50] comprises three main domains for bias evaluation: pre-intervention, during the intervention, and post-intervention. The risk of bias was judged for each domain and sub-domain and classified as low, moderate, high, or no information (Appendix A).

The two authors resolved disagreements through discussion or involving a third author (F.P.). In line with the ROBINS-I tool, the authors considered an NRSI at low risk if judged at low risk of bias for all domains; at moderate risk if judged at moderate risk for at least one domain; at high risk if judged at high risk of bias for at least one domain but not at critical risk of bias in any domain; and at critical risk if judged at critical risk in at least one domain. In addition, we indicated “no information” for an NRSI in case no clear judgment of high or critical risk of bias was possible and in case information about one or more key domains was missing.

## 3. Results

As shown in Appendix A, we retrieved 188 articles and excluded 115 by initial screening of titles and abstracts as not addressing the topics of interest. We included the remaining 73 articles in the final analysis as relevant for the full-text screening. We excluded 39 of them after careful reading: 10/73 were narrative reviews or reviews that did not analyze studies on patients in real-world conditions or therapeutic and/or tolerability outcomes, 26/73 were studies including patient populations different from the target ones, and 3 were studies with only abstracts written in the English language. The remaining 34 articles were eligible to be included in our review.

We further subdivided the 34 studies according to the outcome analyzed regarding effectiveness and tolerability, which were examined based on the type of AP formulation (oral vs. LAI) used to treat enrolled patients, as reported in Appendix A. Thus, the studies reporting the effectiveness of SGAs were sub-grouped into oral SGAs (15/34) and LAI SGAs (19/34) subgroups. Finally, only 11 studies reported data on the tolerability profile of SGAs, namely 3 studies involving oral SGAs and 8 LAI SGAs (Appendix A).

The overall risk of bias was moderate for most non-randomized clinical studies (20/34). Instead, the risk of bias appeared low for one study, with those remaining (13/34) presenting a high risk (Appendix A).

### 3.1. Studies Investigating the Effectiveness of SGAs

All the 34 retrieved studies reported the effectiveness of SGAs in patients with a diagnosis of schizophrenia or related disorders. A total of 15 studies evaluated the effectiveness of SGAs in patients treated with oral formulations and the other 19 in patients treated with LAI SGAs.

SGAs included amisulpride, clozapine, olanzapine, quetiapine, risperidone, paliperidone, ziprasidone, aripiprazole, brexpiprazole, and lurasidone, as a monotherapy or in combination. All the studies emphasized that clozapine was not to be used in combination with other SGAs.

FGAs were prevalently used as an all-drug comparison group and included haloperidol, zuclopenthixol, flupentixol, and sulpiride. In some studies, FGAs were also used in combination therapy with SGAs.

#### 3.1.1. Studies Investigating the Effectiveness of Oral SGAs Treatments

In Table 1, we summarized the results of our literature search on effectiveness outcomes. We described the effectiveness of each treatment and subdivided the 15 studies we analyzed as follows: six studies were on SGAs vs. FGAs, four on olanzapine vs. risperidone, two on ziprasidone not compared with other SGAs or FGAs, one on clozapine vs. other SGAs or FGAs, and one on lurasidone and brexpiprazole, each drug vs. other SGAs.

Most studies evaluated the effectiveness of SGAs vs. FGAs [55,56,57,59,62,65]. Olanzapine, in particular, emerged as an effective treatment option among the atypical agents [51].

Only a few studies directly evaluated the therapeutic effects of SGAs on positive, negative, and affective symptoms [41,42,46,55,56], and none reported antipsychotic effectiveness in disabling cognitive symptoms.

Persistence, adherence, or failure to treatment, as well as the rate of SGAs discontinuation or risk of hospitalization, were analyzed in most studies [52,53,54,55,57,58,59,60,61,62,64,65]. Overall, olanzapine demonstrated superior real-world effectiveness vs. risperidone in relapse-free survival and psychiatric hospitalization [61]. Moreover, switching to clozapine, to risperidone or to olanzapine oral monotherapy was also associated with significantly better persistence in treatment [62]. In addition, Hatta et al. (2022) [65] suggested that switching to LAIs or APs polytherapy might be more likely associated with a low treatment failure rate [65]. Clozapine, as well, was associated with the lowest rates of treatment failure and more marked effects vs. other SGAs in reducing the period of hospitalization [58].

Refs. [52,53] reported on ziprasidone effectiveness, concluding that the improvement in PANSS factors and GAF scores was significant but associated with a discontinued treatment for any cause in more than 50% of patients [52,53]. Discontinuation due to lack of clinical effectiveness was linked more to patients’ perceptions (25.7%) than to physicians’ conclusions (8.6%). However, both studies did not include a control group for comparison. Differently, the study by [60] reported that patients treated with lurasidone demonstrated greater adherence when compared to patients treated with other SGAs [60]. Finally, when brexpiprazole treatment was examined compared to other SGAs, it was found to be associated with fewer psychiatric hospitalizations per year than paliperidone and quetiapine. No significant differences in other efficacy measures emerged between patients treated with brexpiprazole and those with other SGAs [63].

The overall risk of bias for most non-randomized clinical studies reporting the effectiveness of oral SGAs was moderate (11/15).

#### 3.1.2. Studies Investigating the Effectiveness of LAI SGAs Treatments

Most studies reporting on LAI APs treatments included patients that had been previously treated with oral FGAs or SGAs or switched from one LAI FGA/SGA to another LAI SGA treatment (Table 2). Some studies described patients previously treated with the corresponding oral formulation and then shifting to LAI therapy. Furthermore, most studies included patients treated with once-monthly paliperidone palmitate (PP1M) and aripiprazole LAI (Table 2). On the other hand, only a few studies compared the effectiveness of LAI SGAs vs. LAI FGAs, or oral FGAs/SGAs vs. LAI FGAs/SGAs, or oral SGAs vs. LAI SGAs [66].

Finally, only the study by [69] presented results on the effectiveness of LAI risperidone in a retrospective cohort study vs. all-oral SGAs and FGAs and vs. oral risperidone [69]. All the studies, including patients treated with PP1M, reported significant improvements in subjective well-being and global satisfaction, and improved personal and social performance [59,65,67,68,70,71,72,74,75,76,78,79,82]. Furthermore, functionality improvement was more remarkable in patients with a disease duration of 5 years or less [75]. Finally, in a longitudinal prospective study, Ref. [79] reported that PP1M and once-monthly aripiprazole LAI improved social and cognitive functioning in patients who had already experienced relief compared with the corresponding oral formulations of SGAs [79]. In addition, a few studies reported that high doses of PP1M (175 mg equivalent/28 days) in patients with severe schizophrenia improved the drug’s effectiveness [71]. Furthermore, when patients receiving doses of PP1M ≥175 mg Eq were compared to patients treated with high doses of risperidone-LAI (dose ≥ 75 mg) or aripiprazole-LAI (dose ≥ 600 mg/month), PP1M showed better clinical effectiveness, besides reducing the risk of hospital admissions and suicide attempts [78].

Additionally, patients enrolled in other studies showed a low dropout rate, reduced acute healthcare use, and significantly improved neurocognitive function after 12 months of treatment with LAI SGAs, besides better effects on positive, negative, and affective symptoms, psychosocial functioning, and quality of life [79,82]. Furthermore, the transition from PP1M to PP3M evidenced a substantial decrease in combined medications and healthcare resource use, and increased adherence [74].

Treatment with once-monthly aripiprazole LAI improved BPRS and CGI-S scores, especially in younger patients (age ≤ 35 years) [71] and was less likely to be associated with discontinuation of treatment when compared with the corresponding oral group or other SGAs [65,72,73,77,79,80,81]. Thus, adherence and the hospitalization rate appeared to be improved. Such a pharmacological pattern indicates the potential for greater clinical stability in patients who initiated aripiprazole LAI than that achieved with their previous treatments [60].

The risk of bias for non-randomized clinical studies reporting the effectiveness of LAI SGAs was almost equally distributed between moderate (9/19) and high (10/19) risk.

### 3.2. Studies Investigating Tolerability of Oral or LAI SGAs

Table 3 illustrates real-world studies investigating the tolerability of oral or LAI SGAs in patients with schizophrenia and related disorders.

Ref. [51] sustained that the number of patients presenting side effects when treated with SGAs (amisulpride, clozapine, olanzapine, quetiapine, and risperidone) was in the range of 25–63.3%. However, the authors did not specify the secondary or adverse effects reported by patients [51]. On the other hand, among all patients who completed treatment with oral ziprasidone monotherapy, the most common adverse events from baseline to endpoint were mild/moderate [53].

Ref. [83] reported that most frequent adverse effects in patients treated with clozapine (N = 2835) were observed in the three months following treatment start [83]. However, higher percentages of all adverse effects appeared in the first month of clozapine therapy. Furthermore, the data analysis showed a significant negative association between most adverse drug reactions and smoking status, hospital admission conditions, gender, ethnicity, and age of the included patients [83].

Among studies on the tolerability profile of LAI SGAs, six out of eight studies included patients under PP1M treatment. Most studies evidenced treatment-related adverse effects occurring in ≥5% of patients and mainly represented by pain in the injection site (2.3%), insomnia (8.6%), anxiety (6.7%), psychotic disorder (6.1%), headache (5.6%), weight increase (11.9%), and akathisia (11.1%) [67,68,70,71,73,75,78]. Instead, at each assessment point, no significant differences arose in blood pressure, glycemia, triglycerides, total cholesterol, and HDL cholesterol mean scores [71,84]. In the total patient population, 5.7% had a potentially prolactin-related adverse effect (prolactin elevation, amenorrhea/menstrual irregularity in female patients, galactorrhea, gynecomastia, erectile dysfunction, and decreased general sexual function in males) which greatly affected compliance to treatment [67,68,70,75,78,84]. Furthermore, when used at higher doses than standard ones (≥175 mg Eq), PP1M showed a good tolerability profile [71].

The overall risk of bias for non-randomized clinical studies reporting the tolerability results of oral or LAI SGAs was moderate (6/11) and high (5/11).

## 4. Discussion

Overall, the real-world studies analyzed in the present review evidenced that SGAs effectiveness proved superior vs. FGAs, in terms of relapse-free survival, discontinuation rate, and psychiatric hospitalization rate. Furthermore, SGAs were likely superior to FGAs for treating negative symptoms.

On the contrary, RCT results showed that SGAs did not appear to have a better efficacy on negative symptoms than FGAs, although some other studies showed a good efficacy associated with a favorable side-effect profile [85,86,87]. The CATIE study evidenced that all APs had limitations. Therefore, 74% of patients discontinued their randomized treatment over 18 months due to inefficacy or intolerable side effects. Additionally, SGAs differed neither from each other nor from perphenazine (an FGA) concerning effectiveness or EPS. Several studies included in the present review compared SGAs prevalently to haloperidol, which has an increased propensity to cause drug-induced EPS. Accordingly, there was no evidence that SGAs were better for negative symptoms and cognitive deficits. Individual drugs differed in specific side effects. Olanzapine, for example, proved to be the most effective concerning discontinuation rate (64%), although causing the highest side-effect burden [26].

Furthermore, from studies examined in the present review, LAI APs appeared as the pharmacologic treatments with the highest prevention rates of relapse in patients with schizophrenia and related disorders. The risk of psychiatric rehospitalization was the lowest during monotherapy with once-monthly paliperidone LAI, zuclopenthixol, perphenazine, and olanzapine compared with no use of APs [44]. In addition, all LAI APs appeared to be associated with a lower risk of rehospitalization also when compared with the equivalent oral formulations (i.e., oral olanzapine) [44]. Switching from oral SGAs or FGAs to LAIs or APs polytherapy in early non-responders appeared beneficial for preventing treatment failure in hospitalized patients with acute schizophrenia [46,65]. Better relapse prevention and clinical stability were achieved by switching from one LAI to another when deemed necessary [65]. Finally, a more favorable tolerability profile was described in patients switching from oral aripiprazole to aripiprazole LAI [73]. Side effects, such as weight gain, EPS, those related to hyperprolactinemia, and sexual dysfunction, rarely emerged [71,78]. Overall, EPS were present only in patients > 35 years diagnosed with schizophrenia more than 5 years before.

Different long-term SGAs efficacy and tolerability patterns emerged prevalently from meta-analyses of RCTs, which indicated that: (1) regarding all-cause discontinuation, clozapine, olanzapine, and risperidone were significantly superior to several other SGAs, while quetiapine was inferior to several other SGAs [88,89]; (2) as to psychopathology, clozapine and olanzapine were superior to several other SGAs, while quetiapine and ziprasidone were inferior to several other SGAs [90,91]; (3) regarding intolerability-related discontinuation, risperidone was superior and clozapine inferior to several other SGAs [20,92,93]. Concerning weight gain, olanzapine was worse than all the other compared non-clozapine SGAs, and risperidone was significantly worse than several other SGAs. Regarding prolactin increase, risperidone and amisulpride were significantly worse than several other SGAs. Regarding parkinsonism, olanzapine was superior to risperidone, without significant differences about akathisia. Concerning sedation and somnolence, clozapine and quetiapine were significantly worse than a few other SGAs.

However, the apparent improvement in key clinical domains (e.g., negative symptoms) reported by meta-analyses may be largely attributable to improvements in a related clinical domain, such as positive symptoms or fewer AP-related side effects (e.g., EPS), a problem often referred to as pseudospecificity [94].

Our analysis evidenced that SGAs therapy persistence and adherence to treatment were higher than with FGAs. Furthermore, some studies concluded that switching to LAIs or APs polytherapy was associated with a lower treatment failure. In addition, general functionality, subjective well-being, global satisfaction, and improved personal and social performance were reported in patients treated with LAI formulations of SGAs (namely, PP1M and once-monthly aripiprazole LAI) when compared with the corresponding oral formulations.

Clozapine, as well, was associated with the lowest rates of treatment failure and greater efficacy vs. the other SGAs, despite being administered exclusively for intolerant and/or non-responder patients and presenting neurocognitive compromise (mainly reduced performance on attention and memory), plus an unfavorable metabolic and hematological adverse-event profile [83,95]. In the 99% of patients entering CATIE phase 2, clozapine also emerged as significantly more effective than the other SGAs, with a median time to discontinuation of 10 months, twice the length of the following best AP, namely olanzapine [96]. Thus, in both CATIE and CUtLASS studies, SGAs were not found to be more effective (except for olanzapine in CATIE) and did not produce measurably fewer EPS overall. Furthermore, clozapine was the most effective for treatment-resistant patients [26,27].

Among the real-world studies we analyzed, only a few reported on new SGAs, such as lurasidone and brexpiprazole. However, patients treated with lurasidone displayed greater adherence when compared to patients treated with other SGAs [60]. Furthermore, one study analyzed the efficacy of brexpiprazole, and no significant differences emerged when treated patients were compared with those treated with other SGAs [63].

No real-world studies on the effectiveness and tolerability outcomes of patients treated with cariprazine were retrieved by our literature search, although the FDA had approved the drug in 2015.

Most studies selected in this literature review present a few methodological limitations relating to the standard use of medical data from insurance companies, patient registries, administrative and healthcare claims database. Such limitations include no verification of the psychiatric diagnosis and treatments received, high polypharmacy rates, limited knowledge of earlier treatment conditions, and emerging side effects. Furthermore, these studies typically do not present measures of laboratory biological parameters, relying on surrogate markers for the presence of a disease (i.e., for diabetes, the prescription of a hypoglycemic agent, or an ICD code for diabetes). Furthermore, the heterogeneity of the studies conducted in different populations over several decades will likely introduce relevant biases. One of the significant limitations of some studies was the limited or absent control over the data collection quality, which reduced the internal validity of the results. Other potential biases may result from unmeasured confounders and insufficient statistical adjustment of confounders. In this respect, retrospective study data do not meet the criteria of reliability and accuracy required by the methodological rigor of RCTs.

## 5. Conclusions

The present review evidenced that SGAs demonstrated superior effectiveness over FGAs in relapse-free survival and psychiatric hospitalization rate and for treating negative symptoms, while no clear evidence emerged regarding the effectiveness on cognitive deficits. In addition, persistence and adherence to therapy were higher with SGAs than FGAs. Most studies concluded that switching to LAIs was significantly associated with a low treatment failure rate than monotherapy with oral SGAs. Significant improvements in general functionality, subjective well-being, and global satisfaction, besides improved personal and social performance, were reported in some studies on patients treated with LAI SGAs. Furthermore, considering safety and tolerability, our literature review suggests that in adult patients with schizophrenia and related disorders, there may be a lower association of weight gain and adverse metabolic effects with ziprasidone, aripiprazole, and some FGAs compared with olanzapine, clozapine, quetiapine, and risperidone.

Finally, it is crucial for the clinicians to be familiar with the various therapeutic options, not neglecting the old medications, which are still in use with acceptable effectiveness.

## Figures and Tables

**Table 1 jcm-11-04530-t001:** Real-world population-based studies investigating the effectiveness of oral SGAs in patients with schizophrenia and related disorders.

Authors, Year of Publication, Country of Study	Type of Study	No. Included Patients, Target Population	Duration of Follow-Up	Outcome Measures of Effectiveness	Treatment Arms	Results
Taylor et al., 2005 UK [51]	Prospective comparative outcome study, no pharmaceutical industry sponsorship	373 In- and out- patients recruited in 2022	6 months	CGI, positive and negative psychotic symptoms, quality of life.	SGAs treatment groups: Ami, Clo, Ola, Que, Ris	Clinical effectiveness: all SGAs produced similar out-comes; Ola and Ris significantly reduced all ratings at 6 months vs. other SGAs.
Ritsner et al., 2007 Israel [52]	Open-label, observational study, funded by Pfizer Pharmaceuticals Israel	70 patients recruited from 2004 to 2006	1 year	Q-LES-Q, severity of symptoms, distress level	Zipra flexible dosage regimen (40–160 mg/day).	Dropout rate: 54.3% Satisfaction with general activity: increased from month 6 onwards. Severity of clinical symptoms and emotional distress: moderate improvements
Ratner et al., 2007 Israel [53]	Open-label, observational trial, funded by Pfizer Pharmaceuticals Israel	70 patients previously treated with FGAs or other SGAs, recruited from 2004 to 2006	1 year	PANSS, CGI-S, and GAF scales; Rate and mean time of discontinuation treatment.	Zipra flexible-dose monotherapy	All PANSS factors and GAF scores: improved (*p* < 0.05). Effect sizes for changes: moderate from baseline to endpoint: PANSS negative (d = 0.58), positive and activation (for both d = 0.64), dysphoric mood (d = 0.54), autistic preoccupations (d = 55) factors, and general functioning (d = 0.78). Discontinuation treatment: 54.3%; Mean time to discontinuation: 4.4 ± 2.7 months.
Kilzieh et al., 2008 USA [54]	Retrospective study, funded by Eli Lilly	495 patients recruited from1999 to 2000	2 years	Medication discontinuation	Ola vs. Ris	Discontinuation rates: lower for Ola (70%) than Ris (76%) (*p* = 0.12). Median time to discontinuation: longer for Ola (150 days) than Ris (90 days) (*p* = 0.04). Self-discontinuation: no significant difference between Ola (50%) and Ris (46%). Switching rate: more likely to occur in Ris (30%) than Ola (20%) group.
Cortesi et al., 2013 Italy [55]	Longitudinal, retrospective/prospective multicenter cohort study (COMETA), funded by Janssen-Cilag Italy SpA	637 patients enrolled from 2006 to 2007 in 86 mental health centers	mean 14.4 (3.0–17.9) months	PANSS, CGI-S, GAF scales; Persistence, compliance, costs and HRQoL	SGAs, FGAs, and SGAs +FGAs vs. untreated patients.	Relapse rate: 17.1% of patients. Switching rate: 13.4% of SGAs treated patients switched to FGAs, combined SGAs and FGAs, or no treatment. Overall, 22.9% of the cohort switched to another class of drugs at least once, 11% at least twice, and 1.3% four or five times. Persistence on treatment: higher with SGAs than FGAs; on average, 402.8 days for SGAs, 263.0 days for FGAs. The naïve patients had an improvement higher than the non-naïve patients on HRQoL (SF-36 PCS and MCS scores).
Novick et al., 2016 UK [56]	Prospective study (SOHO study), no pharmaceutical industry sponsorship	3712 patients from Europe, Latin America, North Africa, Middle East and East Asia, enrolled from 2000 to 2001	3 years	CGI-SCH negative and positive symptoms. Improvement in social functioning.	Oral Ola vs. other oral SGAs (Ris, Que, Ami, Clo, other SGAs) vs. FGAs.	Negative symptoms and social functioning: SGAs likely superior to FGAs; Overall, negative and depressive symptoms: Ola more effective. Rates of treatment discontinuation: at 36 months lower in Ola-treated patients (38.4%)
Vanasse et al., 2016 Canada [57]	Retrospective cohort study, no pharmaceutical industry sponsorship	18,869 patients enrolled from 1998 to 2005	2 years	Risk of AP discontinuation, switch/add-on AP treatment; combination discontinuation and switching of APs.	All FGAs as single category vs. SGAs (Ola, Ris, Que, Clo)	Risk of stopping or changing medication: lower for Clo, Que, Ola, and Ris vs. FGAs. Clo was the most effective SGA, and Que was the least.
Misawa et al., 2017 Tokyo, Japan [58]	Retrospective mirror-image study, chart review study, no pharmaceutical industry sponsorship	35 patients treated with Clo before 2015, who had taken any SGAs for at least 1 year before initiating Clo.	1 year	Hospitalization and seclusion rates.	Clo vs. other SGAs (Ola, Ris, Ari, Que, Blon, Pali, Peros, PP1M) or FGAs (oral or LAI formulation)	Length of hospitalization: Clo more effective than other SGAs (median value for SGA 110 days and 80 days for Clo; *p* = 0.054). Total days of seclusion: no days during the Clo phase (*p* < 0.001) compared to SGAs (5 days). The number of patients who were secluded at least once was significantly lower (*p* = 0.005) in the Clo phase (*n* = 5; 17.2%) than in the SGA phase (*n* = 17; 58.2%).
Tiihonen et al., 2017 Sweden [59]	Prospectively, nationwide study, funded by Janssen-Cilag	29,823 Patients diagnosed with schizophrenia from 2006 to 2013	Mean 5.7 years (median, 6.9 years).	Risk of rehospitalization; treatment failure	Oral FGAs (Flup, Halo, Perph, and Zuclo) vs. oral SGAs (Ari, Clo, and Ola).	Risk of psychiatric rehospitalization: lowest with Clo monotherapy vs. no use of APs; highest risk with oral Fluph, Que, and Perph; Clo associated with the lowest rates vs. oral Ola.
Rajagopalan et al., 2017 USA [60]	Retrospective study, funded by Sunovion Pharmaceuticals Inc.	1413 patients with a first SGAs prescription claim from 2009 to 2012	6 months	Adherence/medication possession; ratio/proportion of adherent/non-adherent patients; discontinuation rate/mean time to discontinuation	Lura vs. other oral SGAs (Ari, Ola, Que, Ris, and Zipra)	Discontinuation rate: lower for Lura vs. all other SGAs (49.3% vs. 62.3–68.3%, all *p* < 0.05), except for Ris (*p* < 0.05). Mean time to discontinuation: longer for Lura than for other SGAs. Adherence: greater for Lura vs. other SGAs.
Zhang et al., 2019 Shanghai, China [61]	Prospective, multicenter study (SALT-C study), no pharmaceutical industry sponsorship	373 patients receiving Ola, Ris, or Ari monotherapy at least 13 weeks after the baseline visit, recruited from 2011 to 2014	Follow-up times: 13, 26, 52, 78, 104, 130, and 156 weeks after baseline	Discontinuation rate; changes in social functioning (PSP score)	Three SGAs (Ola, Ris, and Ari) as monotherapy.	All-cause discontinuation rate: higher for Ris, lower for Ola and Ari before 24 months but higher in patients taking Ari after 24 months. PSP improvement: maximum value of 80.3% at weeks 56.7 after treatment with Ola, 68.2% at weeks 29.2 with Ris, and 23.9% at weeks 36.8 with Ari.
Stam et al., 2020 The Netherlands [62]	Nationwide pharmacy drug dispensing database; prescription data from 1996 to 2017 from ~ 60 community pharmacies, no pharmaceutical industry sponsorship	321 patients previously treated with Clo for ≥ 90 days, then discontinued due to undefined reasons, recruited from 1996 to 2017	Analysis of database prescriptions from 60 community pharmacies	Persistence time, discontinuation rate in patients stopping Clo	SGAs (Clo, Ola, Que, Ris, and Ari) or FGAs (Halo, Zuclo, Flu, and Sulp) in monotherapy or in combination therapy. LAI therapy included only PP1M or Zuclo LAI	Persistence time: SGAs better than FGAs; restarting Clo or switching to Ris or Ola significantly better than other APs.
Yan et al., 2020 USA [63]	Retrospective cohort study, funded by Otsuka and Lundbeck	6254 patients identified as having at least one claim for either Brex or another oral SGAs, recruited from 2015 to 2016	12 months	Risk of psychiatric inpatient hospitalization rate	Brex vs. other oral SGAs (Zipra, Pali, Lura, Ari, Que, Ola, Ris)	Psychiatric hospitalizations rate/year: Pali and Que users worse than Brex users. No significant differences emerged among other SGAs users.
Barbosa et al., 2021 Brazil [64]	Open, non-concurrent, paired and nationwide cohort study, no pharmaceutical industry sponsorship	3416 patients, 1708 treated with Ola, 1708 with Ris, recruited from 2000 to 2015	15 years	Discontinuation treatment	Ola vs. Ris in monotherapy or in combination therapy with other SGAs (including Clo)	Discontinuation rate: 84.4% of total patients, 82.1% of Ola treated patients, and 86.8% for those prescribed Ris; Median time to discontinuation: overall 63 months, Ola 66 months, and Ris 59 months; Relapse-free survival and psychiatric hospitalization: Ola better than Ris (HR = 1.22; 95% CI = 0.99–1.51; *p* = 0.06).
Hatta et al., 2022 Japan [65]	Multicenter, prospective, cohort study, no pharmaceutical industry sponsorship	1011 patients acutely hospitalized from 2019 to 2021	1 year after discharge	CGI-S score, PANSS-8 derived from PANSS-30; Risk of treatment failure	SGAs (Pal, Ola, Ris, Ari, Brex, Blon, Que) or FGAs (Halo, Fluph) in monotherapy or polytherapy.	Treatment failure: 588 patients, due to rehospitalization (513 patients), discontinuation (17 patients), death (11 patients), prolonged hospitalization for one year (47 patients); lower risk with combined Ola and Pali, higher risk with combined Ari and Ola. Risk of Switching to LAIs and APs polytherapy: 23.4% (237 patients) during follow-up, 74.3% (176/237) patients during hospitalization.

Q-LES-Q = Quality of Life Enjoyment and Satisfaction Questionnaire; PANSS = Positive and Negative Syndrome Scale; CGI-S = Clinical Global Impression—Severity scale; GAF = Global Assessment of Functioning Scale; AP = Antipsychotic; FGAs = First-Generation Antipsychotics; Chlorpro = Chlorpromazine; Fluph= Fluphenazine; Flup = Flupentixol; Halo = Haloperidol; Perph = Perphenazine; Peros = Perospirone; Zuclo = Zuclopenthixol; Sulp = Sulpiride; SGAs = Second-Generation Antipsychotics; Blo = Blonanserin; Zipra = Ziprasidone; Ami = Amisulpride; Clo = Clozapine; Ola = Olanzapine; Que = Quetiapine; Ris = Risperidone; Pali = Paliperidone; Ari = Aripiprazole; Brex = Brexpiprazole; Lura = Lurasidone; Zote = Zotepine; COMETA = COMpliance, costs and quality of life-clinical experience in antipsychotic therapy; HRQoL = Health-Related Quality of Life; SF-36 PCS and MCS = Physical (PCS) and Mental (MCS) Component Summary scores of SF-36; SOHO = Schizophrenia Outpatient Health Outcomes; SALT-C = Schizophrenia by Atypical Antipsychotic Treatment in China; CGI-SCH = Clinical Global Impressions Severity Scale—Schizophrenia version; PSP = Personal and Social Performance.

**Table 2 jcm-11-04530-t002:** Real-world population-based studies investigating the effectiveness of SGAs LAI formulations in patients with schizophrenia and related disorders.

Authors, Year of Publication, Country of Study	Type of Study	No. Included Patients, Target Population	Duration of Follow-Up	Outcome Measures of Effectiveness	Treatment Arms	Results
Schreiner et al., 2014 21 European countries [67]	Prospective Multicenter study (from 160 sites in 21 countries); sponsored by Janssen-Cilag	593 patients switched from oral APs who received at least 1 dose of PP1M during the study, recruited from 2010 to 2013	6 months	PANSS total score, PANSS subscale scores, PANSS Marder factor scores; CGI-C scores; PSP total score; PSP domain scores; and subjective well-being (SWN-S and TSQM).	PP1M	PANSS total: decreased from 71.5 (14.6) at baseline to 59.7 (18.1) at the endpoint; 64% of patients showed a ≥20% improvement in PANSS total score. CGI-S score: increased from 31.8% of patients to 63.2% of patients rating mildly ill or less Mean personal and social performance total score: improved significantly for all patients from baseline to endpoint (*p* ≤ 0.0001).
Hargarter et al., 2015 21 European countries [68]	Prospective multicenter, open-label study [PALMFlexS], sponsored by Janssen Cilag International NV	149 patients with acute symptoms, switching from oral APs due to lack of efficacy, recruited from 2010 to 2013	6 months	PANSS total score PANSS subscale and Marder factor, CGI-S score, CGI-C score, PSP total score, and four PSP domain scores (socially useful activities, personal and social relationships, self-care, and disturbing and aggressive behavior); Mini-ICF-APP; SWN-S-short form, and TSQM scale.	Patients switching from oral APs to PP1M.	CGI-C: severity significantly decreased; percentage of patients rated markedly ill or worse decreased from 75.1% at baseline to 20.5% at last observation; patients categorized as minimally (26.5%), much (41.3%), or very much (14.3%) improved. SWN-S total score, TSQM global satisfaction score, TSQM satisfaction scores related to medication effectiveness: significant improvements PSP total score: significantly increased from baseline to last observation.
Chan et al., 2015 Taiwan [69]	Retrospective cohort study, supported by grants from the E-Da Hospital	379 patients recruited from 2011 to 2012	12 months	Rehospitalization rate, length of hospital stay, emergency room visits and medical expenditures.	Oral SGAs (Que, Ola, Ami, Zipra, Pali, Clo, Zote) or FGAs (Chlorpro, Sulp, Halo, Fluph) or oral Ris vs. LAI Ris	Hospitalization rate before enrolment: all-oral APs group 32.1%, oral Ris group 35.9%, and LAI Ris group 88.4% (*p* < 0.0001). After a 1-year follow-up: all three groups showed similar rehospitalization rates (all-oral APs group 28.9%, oral Ris group 30.1%, LAI Ris group, 30.2%, *p* > 0.999); Length of hospital stay, and number of emergency room visits during follow-up: LAI Ris reduced the severity of disease more significantly than oral APs and medical expenditures.
Alphs et al., 2015 USA [70]	Randomized, prospective, multicenter study (PRIDE study), funded by Janssen Scientific Affairs LLC.	444 patients recruited from 2010 to 2013	15 months	First treatment failure in patients treated with PP1M vs. daily oral APs; time to first psychiatric hospitalization or arrest/incarceration; functionality measured by PSP; severity of psychopathology by CGI-S; adherence to treatment	PP1M vs. daily oral APs (Ari, Halo, Ola, Pali, Perph, Que, Ris)	First treatment failure: PP1M significantly delay in time vs. oral APs (*p* = 0.011); observed treatment failure rates were 39.8% and 53.7%. Arrest/incarceration and psychiatric hospitalization, most common reasons for treatment failure in the PP1M and oral APs groups (21.2% vs. 29.4% and 8.0% vs. 11.9%). No significant differences in PSP and CGI-S scale scores.
Fernández-Miranda et al., 2017 Spain [71]	Prospective observational study, no pharmaceutical industry sponsorship	30 patients resistant to previous Aps treatment, recruited from 2012 to 2015	3 years	CGI-S, WHO-DAS, CAN, MARS, laboratory tests, weight measurement, treatment discontinuation	32 months with 150 mg Eq PP1M, then on average dose of PP: 228,7 mg Eq/28 days; range between 175 and 400 mEq	CGI-S, WHO-DAS, CAN, and MARS: significant improvements (*p* < 0.05) from baseline to month 6. Discontinuation rate: 2/30 due to lack of effectiveness. Significant decrease in the use of other Aps and other psychiatric medications (*p* < 0.05).
Pilon et al., 2017 USA [72]	Retrospective longitudinal cohort study, funded by Janssen Scientific Affairs, LLC.	24,662 patients from Claims data for Medicaid beneficiaries recruited from 2009 to 2015	12 months	Adherence; persistence; health care resource utilization; Medicaid spending	LAI SGAs (Ari, Ola, Pali, Ris) vs. oral SGAs (Ari, Asena, Ilop, Lura, Ola, Pali, Que, Ris, Zipra)	Adherence and persistence to therapy: increased in PP-LAI patients, whereas Ari-LAI and Ris-LAI patients similar to oral SGAs patients; persistence significantly better for PP1M and Ris-LAI, whereas Ari-LAI was similar to oral SGAs. Health care resource utilization: fewer long-term care admissions, long-term care length of stay, and home care services with LAI-SGAs; mental health institute admissions and visits were significantly more frequent with oral SGAs. Medical costs: SGA-LAIs lower than oral SGAs, but higher pharmacy costs.
Tiihonen et al., 2017 Sweden [59]	Prospective study from nationwide databases, funded by Janssen-Cilag	29,823 patients recruited from 2006 to 2013	Mean 5.7 years (median, 6.9 years).	Time receiving monotherapy; Time receiving any therapy; Risk of rehospitalization; Treatment failure (suicide attempt, discontinuation or switch to other medication, or death)	LAI FGAs (Fluph, Flupent, Halo, Perph, Zuclo) vs. LAI SGAs (Ola, Pali, Ris)	Risk of psychiatric rehospitalization: lowest during monotherapy with PP1M, LAI Zuclo, LAI Perph, and LAI Ola vs. no use of APs and vs. equivalent oral APs (20–30% lower); Relapse prevention: LAI APs highest rates; treatment failure: All LAI APs had the lowest rates vs. oral Ola.
Schöttle et al., 2018 Germany [73]	Multicenter, prospective study, sponsored by Lundbeck GmbH and Otsuka GmbH.	242 patients recruited from 2014 to 2016	6 months	BPRS, CGI-S, and CGI-I	Patients pre-treated with oral Ari vs. transition to LAI Ari 1-monthly	CGI-S score: proportion of patients with high CGI-S scores decreased and with low scores increased significantly (*p* < 0.001); decreased significantly more in patients ≤ 35 years; BPRS scores improved, especially in younger patients ≤ 35 years.
Patel et al., 2019 USA [74]	Retrospective claims-based study, funded by Janssen Scientific Affairs, LLC.	122 Veterans’ Health Administration patients with Schizophrenia, initiating treatment with PP1M between 2015 and 2017	12-month pre- and post-PP3M initiation	Treatment patterns, healthcare resource use, and costs	Pre- and post-PP3M transition: patients treated with PP1M vs. patients transited to PP3M	Outpatient and pharmacy visits: reduced during transition to PP3M. Adherence to treatment: 64.8% (proportion of days covered 80%) in patients treated with PP1M and 61.5% in those treated with PP3M. Healthcare resource use: outcomes pre- and post-PP3M transition showed lower all-cause outpatient (37.5 vs. 31.1, *p* ≤ 0.0001) and pharmacy visits (56.1 vs. 46.7, *p* ≤ 0.0001): substantial decrease also in concomitant medication use (i.e., antidepressants) in patients during the post-PP3M transition.
Devrimci-Ozguven et al., 2019 Turkey [75]	National, multicenter, retrospective, and mirror-image study; no pharmaceutical industry sponsorship	205 patients who presented their first psychotic attack 1 year or more before the initial PP1M injection, recruitment initiated in 2016	12 months	PANSS, CGI-S, BPRS, PSP, and GAF scores	Before vs. after treatment with PP1M.	Relapse and median number of hospitalizations: reduced. Effects on functionality: positive. Rate of patients readmitted to the hospital for relapse: 79.5% vs. 28.9% (*p* < 0.001) with median number of hospitalizations (2 vs. 0, *p* < 0.001) lower during PP1M treatment vs. the period before PP1M treatment. PANSS score: decreased by 20% or more during treatment in 75.7% of patients. Functionality: higher when the disease duration was 5 years or less.
Takàcs et al., 2019 Hungary [76]	Nationwide, longitudinal study, no pharmaceutical industry sponsorship	12,232 patients recruited from 2012 to 2013, followed up to 2015	2 years	All-cause treatment discontinuation	All patients with newly initiated SGAs during the inclusion period: oral SGAs (Ami, Ari, Clo, Ola, Que, Ris, Pali, Zipra) vs. LAI SGAs (Ris, Ola, PP1M).	Persistence on treatment after 1 year: oral APs varied between 17% (oral Ris) and 31% (oral Ola), LAIs between 32% (Ris LAI) and 64% (PP1M). The 2-year data were similarly in favor of LAIs. Median time to discontinuation: in the oral group, between 57 days (Clo) and 121 days (Ola); in the LAI group between 176 and 287 days.
Fagiolini et al., 2019 Italy [77]	Observational, retrospective study, no pharmaceutical industry sponsorship	261 patients who had started LAI Ari (at least one injection) at least 6 months before the inclusion visit, recruited from 2015 to 2017	6 months	CGI-S, evaluation of schizophrenia dimensions (symptoms and clusters of symptoms) assessed by the LDPS and SCI-PSY questionnaire	Patients treated with LAI Ari.	Persistence on treatment: 225 patients (86%) for at least 6 months; all patients with baseline CGI-S of 1 or 2,95% with CGI-S of 3, 86% with CGI-S of 4, 82% with CGI-S of 5, 73% with CGI of 6, and 90% with CGI of 7. LAI Ari continuation rate: higher (86.2%) in patients with: (1) baseline CGI score ≤ 4; (2) LDPS mania score ≤ 5; (3) psychotic spectrum schizoid score ≤ 11.
Fernández-Miranda et al., 2020 Spain [78]	Observational, mirror-image study, no pharmaceutical industry sponsorship.	150 patients resistant to previous APs treatment, recruited from 2014 to 2016	6 years	CGI-S, WHO-DAS, MARS, laboratory tests, weight measurement	60 patients treated with LAI Ris ≥ 75 mg; 60 treated with 75 mg/month PP1M; 30 treated with ≥ 600 mg/month LAI Ari	Clinical effectiveness: global improvement on all the scales. Hospital admissions and suicide attempts: statistically significant decrease
Magliocco et al., 2020 Italy [79]	Longitudinal prospective study, no pharmaceutical industry sponsorship	32 patients previously treated with oral SGAs, recruited from 2016 to 2018	12 months	Cognitive performance: SCWT and ROCF tests; PANSS, QOLS, PSP	PP1M vs. oral Pali LAI Ari vs. oral Ari	Neurocognitive function: improved significantly after 12 months of treatment with SGA LAI. Clinical improvement: on psychotic symptoms, psychosocial functioning, and quality of life, and no differences emerged between PP1M and LAI Ari; Functional recovery, adherence to treatment, dropout rate, further social and cognitive improvements: improved in patients who had already experienced relief when on oral SGA therapy.
Iwata et al., 2020 Japan [80]	Retrospective, observational cohort study based on a claims database, supported by Otsuka Pharmaceutical Co., Ltd.	198 LAI Ari group; 1240 oral Ari group, receiving a prescription from 2015 to 2017	Between 2 and 3 years	Treatment persistence	LAI Ari vs. oral Ari group	Treatment persistence: in LAI Ari-treated patients significantly longer than those treated with oral Ari. Discontinuation treatment: LAI Ari group significantly less likely to discontinue than the oral group (adjusted HR 0.54, 95% confidence interval [CI] 0.43–0.68).
Fernández-Miranda et al., 2021 Spain [66]	Observational, longitudinal study, no pharmaceutical industry sponsorship	688 patients with severe schizophrenia in standard care treatments in mental health units (MHU) and on specific program for people with severe mental illness (SMIP), recruited from 2012 to 2014 and followed between 2015 and 2019	5 years	Treatment discontinuation, hospital admissions, and suicide attempts	LAI-FGAs/LAI-SGAs vs. oral FGAS/SGAs	Adherence to treatment: all LAI-APs achieved higher adherence (*p* < 0.001), fewer relapses (*p* < 0.001) and suicide attempts (*p* < 0.01) than oral APs in severe schizophrenia patients.
Lauriello et al., 2021 USA [81]	Retrospective observational cohort study funded by Alkermes, Inc.	485 who had used APs in the 60 days preceding the index date, recruited from 2015 to 2017	6 months	Treatment patterns, healthcare resource use, costs before and after initiating LAI Ari	Recent AP LAI group vs. recent oral AP vs. neither an LAI nor oral AP (“no recent AP”).	All-cause inpatient admissions: decreased by 22.4%, along with emergency room visits. All-cause inpatient costs: decreased by an average of USD 2836 per patient (*p* < 0.05) in the 6-month follow-up; outpatient pharmacy costs: increased by US $4121 (*p* < 0.05), resulting in no significant difference in overall costs between the pre- and post-treatment periods. Discontinuation rate: 29.0%, 40.0%, and 32.9% in the three study subgroups.
Mahabaleshwarkar et al. (2021) USA [82]	Retrospective mirror-image study, funded by Janssen Scientific Affairs, LLC.	210 in patients with at least one oral APs prescription during the 12-month pre-index period, recruited from 2008 to 2020	12-month pre- and post-index periods	Rate of healthcare use: inpatient, emergency room, and outpatient visits	PP1M treatment	Acute healthcare use: reduced significantly from 61.4% to 20.5%, (*p* ≤ 0.001). A more substantial reduction was observed in patients with a prior relapse vs. the overall cohort.
Hatta et al., 2022 Japan [65]	Multicenter, prospective, cohort study, no pharmaceutical industry sponsorship	1011 patients with acute onset or exacerbation of schizophrenia and other psychotic disorders, recruited from 2019 to 2021 and followed up to March 2021	19 months	Risk of treatment failure	Oral SGAs (Pali, Ola, Ris, Ari, Brex, Blon, Que) or FGAs (Halo, Fluph) in monotherapy or polytherapy (excluded Clo) vs. LAI group (Pali, Ari, Halo, Ris, Fluph).	Treatment failure: low rate (588 patients, 58.2%); rehospitalization (513 patients), discontinued medication (17 patients), death (11 patients), and continued hospitalization for one year (47 patients); lower risk in about 19% of patients treated with LAIs and 17% in those with APs polytherapy, vs. patients treated with oral APs. Switching to LAIs or APs polytherapy (no Clo allowed): in early non-responders, it appeared beneficial for preventing treatment failure in acutely hospitalized patients; Ola combined with Pali was significantly associated with a lower risk of treatment failure than monotherapy.

AP = Antipsychotic; FGAs = First-Generation Antipsychotics; SGAs = Second-Generation Antipsychotics; PANSS = Positive and Negative Syndrome Scale; QOLS = Quality of Life scale; PSP = Personal and Social Performance Scale; SWN-S = Subjective Well-being under Neuroleptics Scale; TSQM = Treatment Satisfaction Questionnaire for Medication; PALMFlexS = Paliperidone Palmitate Flexible Dosing in Schizophrenia; CGI-C = Clinical Global Impression—Change; CGI-S = Clinical Global Impression Severity Scale; BPRS = Brief Psychiatric Rating Scale; GAF = Global Assessment of Function; Mini-ICF-APP = Mini-ICF (International Classification of Functionality, Disability and Health) rating for Activity and Participation Disorders in Psychological Illnesses; PP1M = once-monthly paliperidone palmitate; PRIDE = Paliperidone Palmitate Research in Demonstrating Effectiveness; PP3M = once-every-3-months paliperidone palmitate; LDPS = Lifetime Dimensions of Psychosis Scale; SCI-PSY = Structured Clinical Interview for the Psychotic Spectrum; SCWT = Stroop Color and Word Test; ROCF = Rey–Osterrieth Complex Figure Test.

**Table 3 jcm-11-04530-t003:** Real-world population-based studies investigating the tolerability of oral and/or LAI formulations of SGAs in patients with schizophrenia and related disorders.

Authors (Year of Publication), Country of Study	Type of Study	No. of Analyzed Patients	Duration of Follow-Up	Tolerability Results
Taylor et al. (2005) UK [51]	Prospective comparative outcome study with Ami, Clo, Ola, Que, and Ris., no pharmaceutical industry sponsorship	373 In- and out- patients recruited in 2022	6 months	Rate of side effects: 50% (Ami), 60% (Clo), 25% (Ola), 37.5% (Que), 63.3% (Ris).
Ratner et al. (2007) Israel [53]	Open-labeled, flexible-dose, large-scale, observational trial of oral ziprasidone monotherapy, funded by Pfizer Pharmaceuticals Israel	32/70 completed ziprasidone treatment, recruited from 2004 to 2006	1 year	Vital signs, ECGs, or clinical laboratory variables associated with treatment: no significant changes; ESRS, DSAS, weight, and DAI-30: no significant differences during the three follow-up visits (*p* values ≤ 0.05). Adverse events from baseline to endpoint: mild or moderate fatigue (22–28%), sleep disturbances (12–22%), headache (12–16%), somnolence (16–12%).
Iqbal et al. (2020) UK [83]	Data from de-identified EHRs of three mental health trusts in the UK no pharmaceutical industry sponsorship	2835 selected patients under clozapine treatment from 2007 to 2016	Not applicable	Highest recorded adverse effects: sedation, fatigue, agitation, dizziness, hypersalivation, weight gain, tachycardia, headache, constipation, and confusion in the three months following the treatment start; higher percentages of all adverse effects displayed in the first month of therapy; ADRs’ significant association of gender and ethnicity in 7/33, smoking status in 21/33 and hospital admission in 30/33.
Schreiner et al. (2014) 21 European countries [67]	Prospective, interventional, single-arm, multicenter study, sponsored by Janssen-Cilag	593 non-acute symptomatic patients unsuccessfully treated with oral APs; all patients were treated with flexible-dose PP1M, recruited from 2010 to 2013	6 months	Follow-up side effects: 59.7% of patients experienced at least 1 treatment-related side effect; 93.1% of side effects were rated mild or moderate in intensity; 75.8% of adverse effects resulted in no dosage change. Treatment-related adverse effects occurring in ≥5% of patients: injection site pain (2.3%), insomnia (8.6%), anxiety (6.7%), psychotic disorder (6.1%), and headache (5.6%); 18 patients (3.0%) reported at least one potentially prolactin-related side effect, four (0.7%) hyperprolactinemia, and seven (1.2%) potentially prolactin-related side effects as well as hyperprolactinemia. Mean increase of 0.4 kg/m^2^ (95% CI, 0.3–0.6) in BMI and mean weight change between baseline and endpoint of 1.2 kg (95% CI, 0.7–1.6) in the whole group; 81 patients (15.4%) had a ≥7% increase in weight from baseline to endpoint. No EPS were evidenced in all groups.
Hargarter et al. (2015) 21 European countries [68]	Prospective, multicenter, open-label study [PALMFlexS], sponsored by Janssen Cilag International NV	149 patients treated with PP1M flexible dosing, recruited from 2010 to 2013	6 months	Treatment-related side effects: 63.7% of patients experienced at least one, the majority (89.1%) of which were rated as mild or moderate in intensity and did not result in a PP1M dose change (69.7%). Treatment-related side effects reported in ≥5% of patients: injection site pain (13.7%), insomnia (10.8%), psychotic disorder (10.4%), headache, and anxiety (6.1%). Discontinuation treatment: overall, 19 patients (9.0%) reported one or more adverse effects that led to early termination of treatment; most frequent adverse effects leading to discontinuation were psychotic disorder (*n* = 4, 1.9%), acute episode of schizophrenia (*n* = 2; 0.9%) and amenorrhea (*n* = 2; 0.9%). In the total cohort, 12 patients (5.7%) had a potentially prolactin-related adverse effect, 2 (0.9%) hyperprolactinemia, and 1 (0.5%) both. Adverse effects reported as potentially prolactin-related: amenorrhea (2.4%), galactorrhea (0.5%), erectile dysfunction (1.4%), gynecomastia (0.5%), and sexual dysfunction (1.4%). Overall, 40 patients (22.5%) had a ≥7% increase in body weight.
Alphs et al. (2015) USA [70]	Randomized, prospective, open-label, parallel-group, multicenter study (PRIDE study), funded by Janssen Scientific Affairs LLC.	444 patients under flexible monthly maintenance doses of PP1M within a range of 78–234 mg, recruited from 2010 to 2013	15 months	The five most common treatment-related side effects were: pain in the site of injection (18.6%); insomnia (16.8%); weight increase (11.9%); akathisia (11.1%); and anxiety (10.6%). The incidence of hyperprolactinemia was 23.5%, associated with sexual dysfunctions.
Rosso et al. (2016) Italy [84]	Multicenter prospective observational study, no pharmaceutical industry sponsorship	60 inpatients and outpatients treated with PP1M flexible maintenance dosage within the range of 50 to 150 mg Eq, recruited from 2013 to 2014	12 months	The proportion of patients with MetS did not significantly change at 6 (39.0%) and 12 months (29.5%) of PP1M treatment vs. baseline (33%); no significant variation emerged between MetS individual components at baseline and 6 and 12 months. Among the study completers without MetS at baseline (*n* = 30), only two patients (6.6%) fulfilled MetS criteria at the end of the study period (12 months); among study completers with MetS at baseline (*n* = 14), four patients (28.5%) did not fulfill MetS criteria at the end of the study period. A significant increase in BMI (26.3 ± 6.0 vs. 27.1 ± 4.6, *p* = 0.031) and waist circumference (98.2 ± 17.9 vs. 100.3 ± 15.9, *p* = 0.021) from baseline to endpoint. Weight gain in approximately 15% of patients. Rate of ADR: At least one mild or moderate ADR in 71.3% of patients (at baseline), 88.0% (at 6 months), and 52.1% (at 12 months); at each assessment point, no significant differences were found in blood pressure, glycemia, triglycerides, total cholesterol, and HDL cholesterol mean scores. Hyperprolactinemia: in four patients (6.6%) at baseline, six patients (10.1%) at T1, and six patients (13.6%) at T2; symptomatic in two women that showed amenorrhea.
Fernández-Miranda et al. (2017) Spain [71]	Prospective, observational study, patients resistant to previous oral or LAI FGAs and/or SGAs, no pharmaceutical industry sponsorship	30 patients treated with 150 mg Eq PP1M, then on average dose of PP1M 228,7 mEq/28 days; range between 175 and 400 mEq, recruited from 2012 to 2015	3 years	ADR rate: no patients experienced serious adverse events. Discontinuation rate: only one patient due to metabolic syndrome. General tolerability: significant weight loss (*p* < 0.05), decreased glucose, total cholesterol, triglycerides, PRL levels and EPS
Schöttle et al. (2018) Germany [73]	Multicenter, prospective, non-interventional study, sponsored by Lundbeck GmbH and Otsuka GmbH.	242 patients switching from oral-Ari to Ari-LAI, recruited from 2014 to 2016	6 months	Side effects: weight gain (0.4%), experiencing EPS (2.9%), hyperprolactinemia-related side effects (0%) (such as sexual dysfunction), EPS in patients > 35 years who were diagnosed with schizophrenia more than 5 years before.
Devrimci-Ozguven et al. (2019) Turkey [75]	National, multicenter, retrospective, and mirror-image study with PP1M, no pharmaceutical industry sponsorship	205 patients who presented their first psychotic attack 1 year or more before the initial PP1M injection, recruitment initiated in 2016	12 months	Frequency of adverse events: no significant difference before and during PP1M treatment. Side effects: hyperlipidemia, EPS (Parkinsonism, acute dystonia, and akathisia), sedation, and constipation decreased post-PP1M treatment phase; prolactin elevation, amenorrhea/menstrual irregularity in female patients, and sexual dysfunction increased; body weight increased slightly in both female and male patients.
Fernández-Miranda et al. (2020) Spain [78]	Observational, mirror-image study, no pharmaceutical industry sponsorship	150 patients resistant to previous APs: 60 patients treated with LAI Ris ≥ 75 mg; 60 treated with 75 mg/month PP1M; 30 treated with ≥ 600 mg/month LAI Ari, recruited from 2014 to 2016	6 years	Tolerability profile: good for all LAIs, especially Ari-LAI; two patients discontinued treatment due to side effects (akathisia) with Ari-LAI, five with PP1M (three EPS, one hyper-PRL, and one sedation), nine with Ris-LAI (four EPS, one hyper-PRL, three sedation, and one hyperlipemia). Discontinuation rate: four with Ris-LAI, two with PP1m, and one with Ari-LAI due to a lack of effectiveness.

EPS = Extrapyramidal Symptoms; ESRS = Extrapyramidal Symptom Rating Scale; DSAS = Distress Scale for Adverse Symptoms; ADR = Adverse Drug Reaction; DAI-30= Drug Attitude Inventory; EHR = Electronic Health Records; PP1M = Paliperidone palmitate once-monthly; MetS = Metabolic Syndrome; PRIDE study = Paliperidone Palmitate Research in Demonstrating Effectiveness study.

## Data Availability

All data relevant to this paper are included in the article.

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
