# Peer review of "Second-Generation Antipsychotics’ Effectiveness and Tolerability: A Review of Real-World Studies in Patients with Schizophrenia and Related Disorders"

_jcm, 2022, doi:10.3390/jcm11154530_

Round 1

Reviewer 1 Report

            The manuscript has an interesting objective: to review the literature on SGA AP from naturalistic studies, in the real world. The importance of these studies to somehow support the results of RCTs for routine clinical practice is clear. And the need for them to be methodologically rigorous is indisputable. However, the paper lacks some of the depth expected of papers reviewing this issue, and it has major methodological shortcomings. Because of them, the study could not be considered for publication at this stage. Here it follows my suggestions to improve the manuscript quality:

The title does not correspond to the manuscript, since it speaks of efficacy instead of effectiveness, and of patients with "schizophrenia" and not with "schizophrenia and other related psychotic disorders" as it is done throughout the text. This is not only a problem of the title itself but of the entire study, since throughout the manuscript appears in many cases 'efficacy' when, being a review of naturalistic studies, it is actually 'effectiveness'; On the other hand, the reference to patients diagnosed with schizophrenia is also combined throughout the text with other studies with patients with other psychotic disorders. The use of efficacy and effectiveness indistinctly is also given in the abstract.

The Introduction is very extensive and very general, addressing aspects that are not directly related to the objective of the review. In fact, the first five or six paragraphs could be deleted, and the paragraph beginning with “Over the last 15 years….” should be summarized.

In Methods, the lack of definition of the diagnostic profile of the patients in the review persists. Studies that also include FGA in addition to SGA are also unclear; and, however, those that compare oral to LAI SGA are not clearly defined, although the debate about the LAI superiority in improving adherence and preventing relapses is still ongoing (It would also be interesting to include comparisons between AP at standard and at high doses...). There is a certain confusion about the outcomes, mixing efficacy and effectiveness with tolerability and safety. And the use of the term efficacy instead of effectiveness continues throughout this section. It would be more appropriate for primary outcome to review the risk ratio (RR) for hospitalization or relapse, may be by a random-effects model. And other secondary outcomes classed by relevance to effectiveness and safety. Relapse and (re)hospitalization variables are not clearly defined as one of the outcomes, and they are not used as search terms in the review (this possibly means that in the search for articles some of them, with real-world studies that are relevant, have been lost and they do not appear. In short, the method followed for the review should be explained much better, and especially the quality control parameters of the studies used (e.g., PRISMA, or others...). It seems to be necessary to evaluate the quality of studies in terms of risk of bias varied across study designs (cohort studies, pre–post studies, …)  and within each study design from low to high.

Sections 3 (clinical efficacy) 4 (tolerability) should be sub-headings of the 'Results' section. Section 3 describes the instruments used in the studies evaluated; this could be more appropriate in the 'Method' section (also summarizing it as far as possible). As already mentioned, no specific results are shown regarding “relapses” or “hospitalizations”. In subsection 3.1 (clinical studies reporting….), the last paragraphs would be more appropriate in the 'Discussion' section. In subsection 3.2, the results should be summarized much more, highlighting only the most relevant, since they are extensively described in the tables; and the last paragraphs could go (part of them) to the 'Discussion' section. As for the tables that describe the results (tables 1, 2 and 3), they should be summarized much more, especially in the “evaluated outcomes” and “results” columns. Being so extensive, their consultation becomes cumbersome, and an effort of synthesis must be made so that the relevant information is visually much clearer.

There is no “Discussion” section: the authors go directly from the “Results” section to the “Conclusions” section. In fact, most of this section is actually “Discussion”. In this section 5, the third and fourth paragraphs seem to be really the conclusions. However, they should be explained better and, especially, in a way that is more consistent with the results obtained. And include in them the previously commented specifications in results and discussion. In paragraph 7, only the first sentence is relevant, the rest being very specific to an antipsychotic for which there are also no special results in the review. Finally, the last paragraph is too speculative and not directly related to the objective of the review, so it would be unnecessary. As for the references, an excessive number of them are not very up-to-date (prior to 2010) and, in many cases, they are also not directly related to the subject under review.

Some articles that have not been included and that could be considered by the authors, due to their relevance in terms of recent Real-World studies in patients with schizophrenia treated with antipsychotics, are suggested below:

Oral vs LAI:

Chawla, K., Bell, M., Chawla, B. Long Acting Injectable versus Oral Antipsychotics in Reducing Hospitalization Outcomes in Schizophrenia: A Mirror-Image Study. Int. J. Emerg. Ment. Heal. Hum. Resil. 2017, 19, 112-118. https://doi.org/10.4172/1522-4821.1000377

Pilon, D., Tandon, N., Lafeuille, M.H., Kamstra, R., Emond, B., Lefebvre, P., Joshi, K., 2017. Treatment Patterns, Health Care Resource Utilization, and Spending in Medicaid Beneficiaries Initiating Second-generation Long-acting Injectable Agents Versus Oral Atypical Antipsychotics. Clin. Ther. 2017. https://doi.org/10.1016/j.clinthera.2017.08.008

And especially:

Fernández-Miranda, J.J., Díaz-Fernández, S., López-Muñoz, F. Oral Versus Long-Acting Injectable Antipsychotic Treatment for People With Severe Schizophrenia. J. Nerv. Ment. Dis. 2021, 209, 330–335. https://doi.org/10.1097/NMD.0000000000001299

Kishimoto T, Hagi K, Kurokawa S, Kane JM, Correll CU. Long-acting injectable versus oral antipsychotics for the maintenance treatment of schizophrenia: a systematic review and comparative meta-analysis of randomised, cohort, and pre–post studies. The Lancet Psychiatry 8(suppl 2), 387-404 DOI: 10.1016/S2215-0366(21)00039-0

Author Response

We have attached a PDF file

Reviewer 2 Report

The topic of the article is very important because despite the growing number of studies on atypical antipsychotics the question on their efficacy and safety (in contrast to typical antipsychotics) is still open.

The methodolofy of search and evaluation of the articles is described in details. The methodology of this part of study is close to the standards of writing of systematic reviews which decreases the probability of cherry-picking. 

However the interpretation of resultsrequires some comments to be done.

Aypical anipsychotics are mostly compared to haloperidol which is characterised with high number of side effects. Therefore according to J.A. Lieberman such a comparison may be incorrect and comarison of atypical antypsychotics Vs perfenazin (conducted by J.A. Lieberman and mentioned in my article "The comparative efficacy of therapy with typical and atypical antipsychotics") is prefered.

The article includes a lot of references to the comparison of atypical antipsychotics Vs antipsychotics of thioxanthene group which are not active antipsychotics. This fact should be taken into account while analysing such results.

To summarize the articles mentioned in the review contains some incorrect comparisons of antipsychotics. Therefore the performed analysis does not allow make a conclusion on benefits and disadvatages on atypical or typical antipsychotics. These limitations are the must to be described in the article.

Author Response

We have attached a PDF file

Reviewer 3 Report

Dear authors, I uploaded my comments as a MS Word document. Hope it works!

Author Response

We have attached a PDF file

Round 2

Reviewer 1 Report

Important effort to improve the manuscript! From my view, it is now ready to be published.

Author Response

We wish to thank the reviewer for the helpful suggestions, which enabled us to improve our review.

Reviewer 3 Report

I thank the authors for investigating much additional work to improve the article and for taking up several of my suggestions.

I only have several minor comments to improve the article. Those are in the attached Word-document

Author Response

See attached PDF file
